# Urinary PSA and Serum PSA for Aggressive Prostate Cancer Detection

**DOI:** 10.3390/cancers15030960

**Published:** 2023-02-02

**Authors:** Naseruddin Höti, Tung-Shing Lih, Mingming Dong, Zhen Zhang, Leslie Mangold, Alan W. Partin, Lori J. Sokoll, Qing Kay Li, Hui Zhang

**Affiliations:** 1Department of Pathology, Johns Hopkins University School of Medicine, Baltimore, MD 21287, USA; 2Department of Pathology, University of Maryland Medical Center, Baltimore, MD 21201, USA; 3Department of Oncology, Sidney Kimmel Comprehensive Cancer Center, Johns Hopkins University School of Medicine, Baltimore, MD 21231, USA; 4Department of Urology, The Brady Urological Institute, Johns Hopkins University School of Medicine, Baltimore, MD 21287, USA

**Keywords:** prostate specific antigen, urine, prostate cancer

## Abstract

**Simple Summary:**

Prostate cancer remains one of the leading causes of death in men in the United States. The commonly utilized method of diagnosing prostate cancer is via the digital rectal examination and serum PSA assay, which is usually followed by biopsy of the prostate gland to evaluate the extent of the cancer. While serum PSA assay has been used for many years by urologists across the globe to diagnose prostate cancer, a falsely elevated level due to conditions (BPH or prostatitis) other than cancer could cause the patient to undergo prostate gland biopsy, which is invasive and poses a greater risk of complications. To overcome these issues, we evaluated whether urinary PSA, which is secreted from the prostate gland into the urine could have predictive value in differentiating aggressive prostate cancer from indolent disease. In this study, we analyzed more than 400 samples for serum and urinary PSA, and found that urinary PSA had a higher predictive power in differentiating aggressive prostate cancer, and could serve a better surrogate tumor biomarker in capitulating the tissue milieus for the purpose of detecting aggressive prostate cancer. Furthermore, combining the ratio between serum to urine PSA enhanced the performance of both biomarkers in predicting aggressive prostate diseases. These studies support the role of urinary PSA, in combination with serum, for detecting aggressive prostate cancer.

**Abstract:**

Serum PSA, together with digital rectal examination and imaging of the prostate gland, have remained the gold standard in urological practices for the management of and intervention for prostate cancer. Based on these adopted practices, the limitations of serum PSA in identifying aggressive prostate cancer has led us to evaluate whether urinary PSA levels might have any clinical utility in prostate cancer diagnosis. Utilizing the Access Hybritech PSA assay, we evaluated a total of n = 437 urine specimens from post-DRE prostate cancer patients. In our initial cohort, PSA tests from a total of one hundred and forty-six (n = 146) urine specimens were obtained from patients with aggressive (Gleason Score ≥ 8, n = 76) and non-aggressive (Gleason Score = 6, n = 70) prostate cancer. A second cohort, with a larger set of n = 291 urine samples from patients with aggressive (GS ≥ 7, n = 168) and non-aggressive (GS = 6, n = 123) prostate cancer, was also utilized in our study. Our data demonstrated that patients with aggressive disease had lower levels of urinary PSA compared to the non-aggressive patients, while the serum PSA levels were higher in patients with aggressive prostate disease. The discordance between serum and urine PSA levels was further validated by immuno-histochemistry (IHC) assay in biopsied tumors and in metastatic lesions (n = 62). Our data demonstrated that aggressive prostate cancer was negatively correlated with the PSA in prostate cancer tissues, and, unlike serum PSA, urinary PSA might serve a better surrogate for capitulating tissue milieus to detect aggressive prostate cancer. We further explored the utility of urine PSA as a cancer biomarker, either alone and in combination with serum PSA, and their ratio (serum to urine PSA) to predict disease status. Comparing the AUCs for the urine and serum PSA alone, we found that urinary PSA had a higher predictive power (AUC= 0.732) in detecting aggressive disease. Furthermore, combining the ratios between serum to urine PSA with urine and serum assay enhanced the performance (AUC = 0.811) in predicting aggressive prostate disease. These studies support the role of urinary PSA in combination with serum for detecting aggressive prostate cancer.

## 1. Introduction

Currently, prostate-serum specific antigen (PSA) is the most commonly used biomarker for the detection of prostate cancer [1,2]. The PSA protein, which is not cancer-specific, but a protein specific to prostate tissue [3,4], is produced from the columnar epithelium cells of the prostate tissue, and is known to have serine protease activity [5,6]. PSA has been shown to be produced both in healthy and hyperplasic cells, as well as cancer cells [7]. Several studies have demonstrated lower levels of PSA production in cancerous tissues [8,9]. Based on a risk benefit analysis of PSA for prostate cancer, the US task force has previously recommended against the use of PSA in prostate cancer screening (category D) [10]; however, they later revised those recommendations and suggested the usage of PSA to be an option for physicians and patients to decide upon if the PSA test could be used [11]. A number of new approaches, including multiparametric MRI and genomic targets, have recently emerged to help manage some of the clinical challenges associated with prostate cancers. These assays, however, suffer from an impediment, that is, how to integrate and use them in patient care and whether the additional information they provide justifies the cost-benefit ratios [12].

In this study, we utilized urine specimens from aggressive (GS ≥ 7) and non-aggressive (GS ≤ 6) prostate cancer patients to evaluate whether the urinary PSA that was released directly from the prostate during the DRE examination would be able to differentiate prostate cancer patients with aggressive diseases. While urine is known to show a high degree of variability for proteins [13], the rationale for utilizing urine specimens was based on the anatomical location of the prostate. The urethra, which runs from the bladder, passes through the prostate to collect prostatic fluid along with mature sperm from the vas deferens [14,15,16]. Secondly, unlike prostate biopsies, which are invasive urine samples can be easily acquired in any routine hospital visit or follow-up. We, along with others, have shown the utility of urine-based proteins or metabolites in the development of cancer biomarkers. In a recent study, Di-Minno et al. evaluated two urinary metabolites as biomarkers to predict the radicality of their treatment procedure in patients with Pca. Using the LC-MS approach, the authors measured levels of 8-OHdG and 8-iso-PGF2 α before and after the robotic-assisted radical prostatectomies (RARP). In their study, they demonstrated that levels of 8-OHdG and 8-iso-PGF2 α were enough to predict the radicality or local recurrence of prostate cancer after the RARP [17]. In this study, we utilized urine specimens to evaluate whether the urinary PSA might have predictive value for aggressive prostate disease.

## 2. Methods and Materials

### 2.1. Urine Samples and PSA Assay

Urine samples and their associated clinical information were obtained from 146 PCa patients (Cohort 1) and the 291 PCa patients (Cohort 2), with approval from the Institutional Review Board (IRB protocols # IRB00181703 and NA_00037951/CR00013095) of The Johns Hopkins School of Medicine. The post-digital rectal examination (DRE) urine samples for cohort 1 represented a distinct patient population, with those with Gleason scores six or below (GS ≤ 6) classified as non-aggressive (NAG), and with samples from patients with pathological Gleason scores of eight or above (GS ≥ 8) considered aggressive (AG). A second group of samples, cohort 2, with larger sample size, also included patients with Gleason scores of 7, that is (GS 3 + 4) and (GS 4 + 3), who were classified as aggressive patients (AG) Appendix A.

Urine specimens were analyzed at the EDRN Biomarker Reference Laboratory at Johns Hopkins University, in a blinded fashion, on the Beckman Coulter Access 2 Immunoassay Analyzer (Beckman Coulter, Inc. Brea, CA, USA) for total PSA. The PSA assay utilizes dual monoclonal antibodies in sandwich assay formats with chemiluminescence detection. Urine samples were diluted 1:10 in assay diluent prior to analysis.

### 2.2. Human Metastatic and Non-Metastatic Tissue Samples

For the human tissue blocks not associated with urine samples, the Johns Hopkins Hospital Department of Pathology archives computer was searched for the diagnosis of ‘metastatic prostate cancer’ from 1994 to 2010, and yielded 32 fine needle aspiration (FNA) cases and 30 biopsy/resection cases. The H&E slides and the immunohistochemistry (IHC) results for the PSA protein were reviewed by a board-certified pathologist. Staining of PSA in metastatic tumors and the morphology of the metastatic tumor were correlated with the primary tumor Gleason scores, and with the patients’ clinical information. The most common metastatic sites, in descending order, were the lung (10 cases), bone (10 cases), distant LN (7 cases) and local LN (4 cases), liver (6 cases), pleura (6 cases), bladder (5 cases), adrenal (1 case), and others (13 cases). Among 62 cases, 46 cases had documented Gleason scores of primary tumors. The clinical information of these cases and the Gleason scores of primary tumors are summarized in Appendix A.

### 2.3. IHC Staining for Tissue

Tissue immunohistochemistry (IHC) staining was performed as described previously [18,19,20]. Paraffin-embedded tissue slides were deparaffinized and rehydrated. Antigen retrieval was performed using the Target Antigen Retrieval Solution (DAKO, Santa Clara, CA, USA) by boiling inside the steamer pot for 25–30 min. The endogenous peroxidase was blocked using a 3% hydrogen peroxide solution, followed by a serum blocking solution from the DAKO. Primary antibodies diluted in the antibody dilution buffer (DAKO, CA, USA) were incubated for 1 h at 37 °C before incubating the slides with primary antibody overnight at 4 °C. The secondary antibody ready to use polymer with anti-rabbit (DAKO, CA, USA) was used at room temperature for 30 min, and bound peroxidase was detected using the DAB (DAKO, CA, USA). All IHC slides were counterstained with hematoxylin. For tissue morphological analysis, microscopic images were examined under 10× or 40× magnification using an inverted light microscope (Zeiss, White Plains, NY, USA).

### 2.4. Statistical Analysis

A non-parametric Mann–Whitney U test was utilized to demonstrate differences between the urine and serum PSA, where statistical significance was defined when the *p*-value was ≤0.05. The Pearson correlation between urine and serum PSA was calculated for each Gleason score. The receiver operating characteristic curve (ROC) was generated using 5 repeats of the 10-fold-cross validation to avoid overestimation. The Area Under the Curves (AUC) and the 95% confidence intervals were indicated in ROC plots. The predictive models (logistic regression) with cross-validation were built using caret (version 6.0-85) in R. The ROC curve was generated using pROC (version 1.13), whereas the AUC, along with 95% confidence interval, was obtained via MLeval (version 0.3) in R.

## 3. Results

### 3.1. Urine and Serum PSA in Aggressive and Non-Aggressive Prostate Cancer

In order to measure levels of PSA in urine and to determine the dilution factor that would be necessary to perform the urinary PSA assay, we pooled equal volumes (10 μL) either from the 146 urine samples, the JHU pool; from n = 70 non-aggressive urine sample, the JHU_NAG pool; or from n = 76 aggressive urine samples, the JHU_AG Pool. The pooled urine samples were then diluted into 5, 10, 50, and 500-fold using the PSA dilution buffer (Appendix A). The clinical PSA assay (Access 2 Hybritech PSA Assay) was utilized to measure the total urinary PSA. The 1:10 fold urine dilution displayed values that were within the linear range of the assay, and was selected as the working dilution for the rest of the analysis of the urine samples. The serum PSA values from the n = 146 patients were obtained from the Johns Hopkins patients’ database, and were plotted to show levels of total serum PSA between NAG and AG (Figure 1A). Using the 1:10 dilution factor for urine, we measured levels of urinary PSA in each sample from aggressive (GS ≥ 8 (n = 76)) and non-aggressive (GS ≤ 6 (n = 70)) prostate cancer patients. As shown in Figure 1B, patients that had aggressive disease showed lower levels of total urinary PSA compared to the non-aggressive prostate cancer patients. The median levels of total urinary PSA in aggressive patients were determine to be 81.3 ng/mL compared to 279 ng/mL in non-aggressive patients. To further evaluate the association between serum and urine PSA with aggressive prostate disease, we stratified samples according to their surgical Gleason scores. As shown in Figure 1C,D, serum PSA demonstrated higher levels of PSA with increasing Gleason grade cancers in aggressive prostate cancer patients. Unlike serum, the urinary PSA showed the opposite response, where levels of urinary PSA decreased in higher Gleason grade cancers. The distribution of PSA levels in serum and urine compelled us to evaluate whether serum and urinary PSA might have a negative association in these paired samples or in the same Gleason groups. In order to delineate whether there was an inverse correlation between the urine and serum PSA in the paired cancer samples (Appendix A), we performed the Spearman correlation analysis between the two and found no significant association between the serum and urine specimens (Appendix A).

### 3.2. Urine PSA—A Proxy for PSA Level in Prostate Cancer Tissue

In order to investigate levels of serum and urine PSA, we evaluated the PSA expression in different prostate cancer tissues with metastatic lesions (n = 62) for Gleason 7 and above, as well as four primary prostate adenocarcinomas (GS = 6). The clinical information of the cases and their Gleason scores, along with the presence or absence of PSA expression for these tumors, are summarized in Appendix A. Based on these observations, our data suggest that metastatic prostate cancer tends to lose the expression of PSA protein in cells. Out of the 33 different metastatic tumors that were evaluated for PSA expression, only 19 (58%) tumors were detected to have PSA expression, while 14 (~42%) tumors were totally devoid of PSA expression. Parallel to this observation, the non-aggressive (GS = 6) tumors had higher levels of PSA expression when compared with advanced metastatic tumors (GS ≥ 8) (Figure 2). The tissue expression correlates well with the urinary analysis, which showed lower levels of PSA in aggressive prostate cancer. This lower level or loss of PSA expression in aggressive tumors suggested that urinary PSA might be a better surrogate for tissue PSA levels. These data further suggest that other prostate specific proteins should be evaluated in urine-based analysis to demonstrate their utility in prostate cancer prognosis.

### 3.3. The Combination of Serum and Urine PSA in Predicating Aggressive Prostate Cancer

The need for an aggressive prostate cancer biomarker is driven by the aim to avoid unnecessary biopsies, and, at the same time, the fear of missing or avoiding treatment of curable diseases. Therefore, a great need remains in the milieu of prostate cancer to identify biomarkers that can accurately predict which patients will require treatment interventions. To demonstrate whether total urinary PSA values might add to the utility of serum PSA, we performed a ROC analysis for serum and urine PSA, alone or in combination, to evaluate whether adding both urinary and serum PSA values can enhance the predictive power of the serum PSA in differentiating aggressive from non-aggressive disease. As shown in Figure 2B, serum PSA alone showed a discriminatory power, with an area under the curve (AUC) of 0.695. On the other hand, urinary PSA showed a superior AUC of 0.741, with better discriminatory ability for the prediction of aggressive prostate cancer. The combination of the urine with the serum PSA assay further improved the predictive values (AUC = 0.767) of the serum PSA in detecting aggressive prostate cancer.

### 3.4. Utilizing Serum and Urinary PSA Ratio for Discriminating Aggressive (AG) from Non-Aggressive (NAG) Disease

In order to investigate the relative relationships between the urine and serum PSA with respect to their ability to separate AG from low-risk PCa, we utilized the two-dimensional ellipse graph to evaluate the distribution of both urine and serum PSA. As shown in Figure 3A,B, the majority of the PSA values were centered around the coordinates (7.28, 2.57) within the ellipse. The Log 2 values for urine PSA are shown on the x-axis, while the serum levels are represented on the Y-axis. The absence of discrete separation between the urine and serum PSA groups led us to evaluate the ratios of serum to urine PSA in predicting AG from NAG disease. To demonstrate whether the ratio of serum to urine PSA might be a better predictor of aggressive disease, we initially evaluated 141 urine samples (AG = 71 and NAG-70) from cohort 1 with biopsied Gleason scores (GS ≤ 6 vs. GS ≥ 8). As shown in Figure 3C,D, the serum PSA (*p* = 2.1 × 10^−5^) and the urine PSA (*p =* 2.93 × 10^−7^) showed a statistically significant difference in differentiating AG from the NAG disease. Using the serum to urine PSA ratios from the same patient improved the overall median distribution to differentiate AG vs. NAG prostate cancer patients (*p =* 1.42 × 10^−10^) (Figure 3E). We utilized these observations and conducted the ROC analysis for serum and urine PSA, and found that using ratios for serum to urine yields the highest AUC (0.794) when compared to the urine and serum alone, or their combination (Figure 3F). Similarly, when serum (P1), urine (P2), and their ratios (P3) were combined together, a further improvement was observed with an AUC of (0.811), suggesting the utility of PSA ratios between serum to urine in differentiating AG from NAG disease (Figure 3F). To further stratify the discriminating power of PSA at different cut-offs, we analyzed several AUCs at different PSA levels (<4 ng/mL, <10 ng/mL, <20 ng/mL, or all observed PSA values). As shown in Figure 3G, serum PSA at <4 ng/mL did not perform well (AUC < 0.45); similarly, using a cut-off of <10 ng/mL for serum PSA showed a moderate improvement, with an overall AUC = 0.60, which almost remained identical at serum PSA levels of <20 ng/mL. On the other hand, urinary PSA levels outperformed the serum PSA, especially at PSA levels lower than 4 ng/mL. Using the ratios of serum to urine PSA showed the best overall performance at every single cut-off, with an AUC falling between 0.75 to 0.80 and a *p*-value (comparing a marker against the null of AUC = 0.5) of <0.05 (Figure 3G). The difference between PSA ratio and serum PSA across different serum PSA levels was clinical meaningful when directly comparing the AUCs of the two markers, with *p*-value < 0.1 or even less (<4 ng/mL: *p* = 0.0365, <10 ng/mL: *p* = 0.0913, <20 ng/mL: *p* = 0.0153, All: *p* = 0.0391).

### 3.5. Ratios of Serum to Urine PSA Levels in Predicting the Biopsy and Radical Prostatectomy-Based Gleason Outcomes

To further validate the utility of urinary PSA in complementing the serum PSA assay, we explored another large cohort (n = 291) of urine samples from prostate cancer patients with known Gleason scores (≤6, 7, ≥8). Using the Beckman Coulter Access 2 Immunoassay analyzer the total urinary PSA levels were measured and plotted against serum PSA using two-dimensional graphs (Figure 4A–D) which showed no significant overlap or inverse relationship. The central values for the urinary PSA in biopsied (Bx) samples (GS ≤ 6 and ≥8) were detected at the coordinates of 7.63 and 2.61 (Figure 4A), while the radical retropubic prostatectomy (RRP)-based Gleason (GS ≤ 6 and GS ≥ 8) fell at the coordinates of 7.4 and 2.61 (Figure 4B). The PSA (urine and serum) values for all Gleason grades (GS ≤ 6, GS7 (3 + 4 and 4 + 3) and GS ≥ 8) for the biopsy and radical prostatectomy (7.46, 2.63), and (7.46, 2.63), respectively. By evaluating the AG vs. NAG serum or urine samples that had the biopsy Gleason scores available for the four groups (GS6, GS7(3 + 4) or G7 (4 + 3), GS ≥ 8), we found a significant difference between the two AG vs. NAG groups, with *p*-values of (*p* = 1.03 × 10^−4^) and (*p* = 3.59 × 10^−5^), respectively (Figure 4E,F). To further confirm whether PSA ratios between serum to urine might be superior in discriminating the two groups, we analyzed the ratio of serum to urinary PSA in samples with known biopsied Gleason scores and found that utilizing ratios could improve the overall *p*-value (*p* = 3.71 × 10^−7^) in identifying AG disease (Figure 4G). Samples stratified further across different Gleason grades (≤6, GS7 (3 + 4 or 4 + 3) and GS ≥ 8) were also statistically analyzed (Figure H–J). As shown in Figure 4H, serum PSA in biopsied Gleason GS ≤ 6 vs. GS7 (3 + 4) patients failed to show any statistical differences (*p* =0.224); on the other hand, urine PSA, when compared between GS ≤ 6 vs. GS7 (3 + 4), demonstrated a statistical significance of *p* = 0.013 (Figure 4I). These assays, however, complemented one another when ratios between serum to urine PSA were considered (Figure 4J). A similar evaluation of serum and urine PSA was performed using the pathological Gleason scores and showed similar results, suggesting that ratios for PSA levels between serum and urine from the same patient might be a better approach to overcome the limitations of both assays (Appendix A).

### 3.6. Evaluating the Performance of Serum to Urine Alone and Their Ratios across Different Gleason Scores 

Using the PSA data from the 291 urine and serum samples, we further explored the utility of urine and serum PSA alone and their ratios in predicating the AG disease. A repeated (five times) 10-fold cross validation logistic regression model was used to generate the ROC curves between the PSA values from GS ≤ 6 vs. GS ≥ 8, for both the biopsy and the radical prostatectomy specimens. As shown in the Figure 5A, serum PSA with (Bx) Gleason scores had an AUC = 0.678, compared to AUC of 0.741 when pathological RRP GS were considered. On the other hand, urinary PSA had a slightly better performance (AUC = 0.682) than serum PSA, but only in specimens with biopsy (Bx) GS and without pathological (RRP) GS (AUC = 0.683). However, when ratios of the serum to urine PSA among the grouped samples were considered, a higher AUC was obtained for both biopsy (Bx) (AUC = 0.719) and pathological (RRP) GS (AUC = 0.776). We also evaluated the AUCs for their use in combinatory approaches with urine and serum; however, no significant improvements from the AUC between the ratio for serum and urine alone were achieved (Figure 5B). These confirmative data in the second cohort (n = 291, which also comprised the GS7 (3 + 4 and 4 + 3), once again supported the utility of the serum to urine PSA ratios in clinical practices, which could be considered to differentiate AG from NAG prostate disease.

## 4. Discussion

Serum PSA has helped clinicians for decades in assessing treatment responses [21,22]. However, using serum PSA as a screening method to detect prostate cancer early in the course of a disease is controversial [23,24,25,26]. A number of factors affect the PSA levels, including, but not limited to, prostate inflammation [27,28], benign prostate hyperplasia (BPH) [21,27,29], ejaculation [30,31], urinary retention [32], age [33], and AR targeting medications and their effects on autophagy [34,35]. To overcome these challenges, several assays based on the PSA modifications were proposed to refine the interpretation of elevated PSA. These assays include the use of PSA density, PSA velocity, PSA isoforms to improve the diagnostic accuracy for prostate cancer [2]. 

Over the years, it has become clear that serum PSA offers diagnostic advantages in identifying cancer at or above 10 ng/mL. However, assays to detect prostate cancer earlier in the course of the disease and with greater accuracies are required to detect aggressive tumors in patients for whom treatment might save their lives. With the aim to harness the diagnostic power of PSA, we evaluated levels of urinary PSA in two different cohorts (n = 146, n = 291). These urine samples were acquired through the Johns Hopkins Hospital from aggressive and non-aggressive prostate cancer patients. Interestingly, in our interim analysis of the 146 patients (GS ≤ 6 and GS ≥ 8), urinary PSA was found to be a better predictor than serum PSA in detecting aggressive disease (AUC 0.74 vs. 0.69). A combination of both urinary and serum PSA moderately enhanced the predictive power (AUC = 0.767) of the assay in detecting aggressive prostate cancer. Currently, urine-based biomarker discovery is gaining attention for many urological malignancies, and has become one of the most attractive bio-fluids in clinical proteomics [36,37,38]. Compared with other biological specimens, urine has many advantages for the determination of both diagnostic and prognostic biomarkers [17]. It is collected in a non-invasive manner, without any requirements for specialized laboratory personnel. Furthermore, urine can be obtained in large quantities, it has less complexity in protein composition, and the chance of proteolytic degradation of samples is significantly low. Our data also demonstrated that unlike serum PSA, urinary PSA was more closely associated with tissue level expression. Using the IHC (Figure 2A) to evaluate the PSA expression in primary and metastatic tumors, our data demonstrated that PSA expression was decreased in higher Gleason grade tumors, which is consistent with observations that during aggressive cancer growth, prostate tumors lower dependency on AR regulation [18]. While PSA is a direct downstream target of the activated AR, a lower level of PSA is present in these advanced tumor types [39]. Further verification of these observations will be required to understand the differences between serum and urinary PSA levels.

Using a large cohort (n = 291) and a 10-fold cross validation logistical regression models, we demonstrated that the ratio between serum and urinary PSA could be used to improve the discriminating power of the serum PSA assay in detecting aggressive prostate cancer. To predict the biopsy-based clinical outcomes in the 291 urine samples, we evaluated the performance of serum or urinary PSA, both alone and using their ratios, in groups with biopsy- (Bx) or pathological- (RRP) based GS. Utilizing the ratios of serum to urine from the same patient, higher AUCs were achieved in groups with biopsy-based GS (AUC = 0.719) or pathological GS (AUC = 0.776), suggesting a better performance when using ratios between serum and urine. It will be interesting to combine this assay, which is prostate-specific, to other cancer biomarkers. For example, the T2:ERG and urine PCA3 in a large cohort of prostate cancer patients to evaluate their utility as a novel cancer biomarker panel [40,41].

## 5. Conclusions

Our study demonstrated that urinary PSA could complement the serum PSA, and could be a surrogate biomarker in predicting aggressive prostate cancer. Unlike serum PSA, urinary PSA levels more closely resembled changes associated with tissue levels in prostate cancer tumors. Our study has clinical implications for adapting the use of the ratio of serum to urinary PSA to the detection of aggressive prostate disease. Further studies are warranted to evaluate urinary PSA in larger prostate cancer cohorts, as well as to investigate the effectiveness of the ratio of serum to urine, with different PSA cut-off values, in predicting advanced prostate cancer.

## Figures and Tables

**Figure 1 cancers-15-00960-f001:**
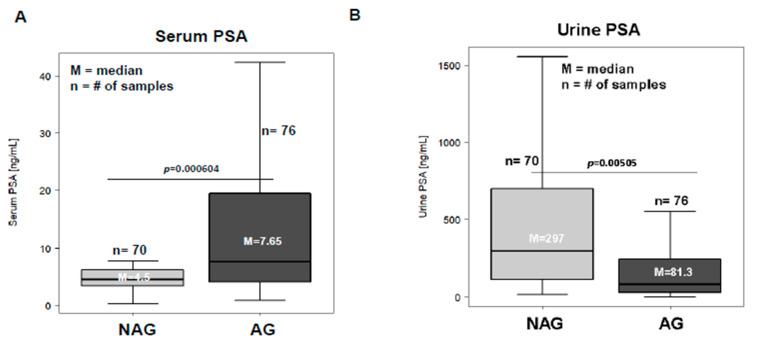
Levels of serum and urine PSA in non-aggressive (GS ≤ 6) and aggressive prostate cancer (GS ≥ 8): Beckman Coulter Access 2 Immunoassay for PSA was performed on n = 146 serum (**A**) and urine (**B**) samples obtained from the same patients. Median levels of serum PSA levels for the NAG patients (n = 70) were found to be 4.6 ng/mL, and for AG patients with (n = 76), they were 7.65 ng/mL (**A**). Urine PSA levels were measured using the same Beckman Coulter Access 2 immunoassay in 146 prostate cancer patients. Median urinary PSA level for NAG patients was 297 ng/mL, compared to the 81.3 ng/mL in AG prostate cancer patients. Significance was defined at *p* ≤ 0.05. Serum and urine PSA from the same patients were analyzed based on different Gleason scores, and the serum PSA (**C**) and urinary PSA (**D**) levels were plotted using box and whisker plots. The *p* values between serum and urinary PSA at different GS are shown in table E. Statistical significance was defined as when the *p* value was *p* ≤ 0.05 (**E**).

**Figure 2 cancers-15-00960-f002:**
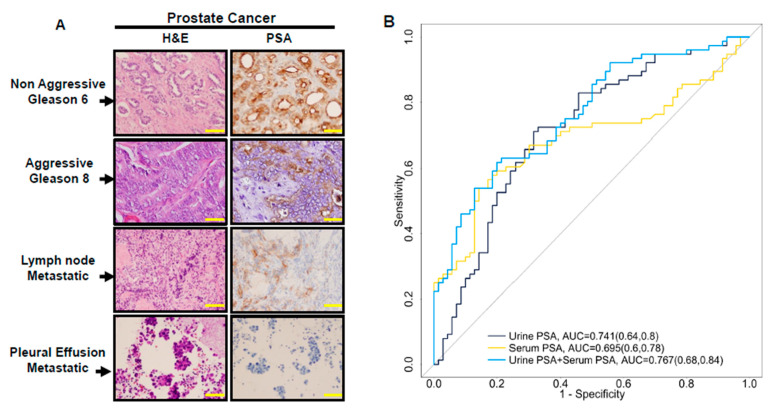
Immuno-histochemistry (IHC) for prostate cancer tissues in aggressive or non-aggressive and metastatic lesions. AUC analysis: the selected 10 × H&E staining and IHC for PSA expression in GS 6 to 8 primary prostate cancer tumors, in lymph node metastasis, and in pleural effusion metastatic tumors were obtained using an inverted light microscope (Zeiss, White Plains, NY, USA). Low staining intensity of PSA expression in aggressive prostate tumors, or loss of expression in some metastatic lesions, were observed (**A**). In ROC analysis, the urinary PSA demonstrated s higher AUC compared to serum PSA in differentiating aggressive from non-aggressive diseases. The combination of serum and urinary PSA further improved the differentiation. A higher ROC curve was obtained when urinary PSA was combined with serum PSA (**B**).

**Figure 3 cancers-15-00960-f003:**
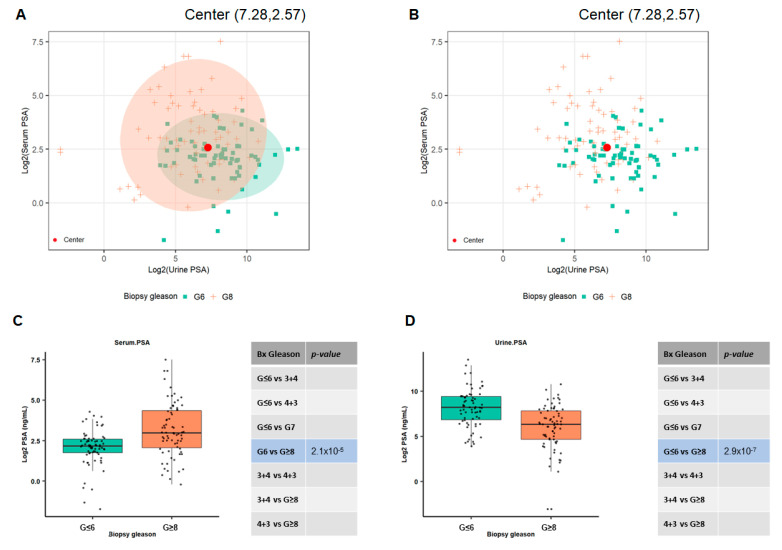
Utilizing urine and serum PSA ratio in differentiating AG from NAG. The two-dimensional plots with ellipse (**A**) and without ellipse (**B**) show the overall distribution of PSA values in urine and serum samples in the (n = 141) cohort 1 for GS ≤ 6 vs. GS ≥ 8 samples. The red dot represents the center distribution point for all the PSA values. The Log 2 values for serum PSA are shown on the *y*-axis, and urine PSA values are given on the *x*-axis (**A,B**). The median values of serum PSA (**C**) and urine PSA (**D**) or their ratios (**E**) in the AG vs. NAG patients were plotted to show the distribution of individual values of PSA. The *p* values for each group are given on each side of the plot. Significance was set at *p* < 0.05. The ROC curve between serum (P1) and urine (P2) or the ratios of serum to urine (P3), as well as the combinations of serum and urine along with the ratios, demonstrate the utility of serum to urine PSA. Numbers in parentheses represent the 95% (CI) confidence intervals (**F**). The performance of urine PSA and PSA ratio at different serum PSA levels were investigated using only the samples with serum PSA levels below the cutoff (4 ng/mL, < 10 ng/mL, <20 ng/mL). The AUCs obtained from using all the samples, regardless of serum PSA level, were included in the plot (i.e., All) as well. The *p*-values were from comparing a marker against the null of AUC = 0.5 (**G**).

**Figure 4 cancers-15-00960-f004:**
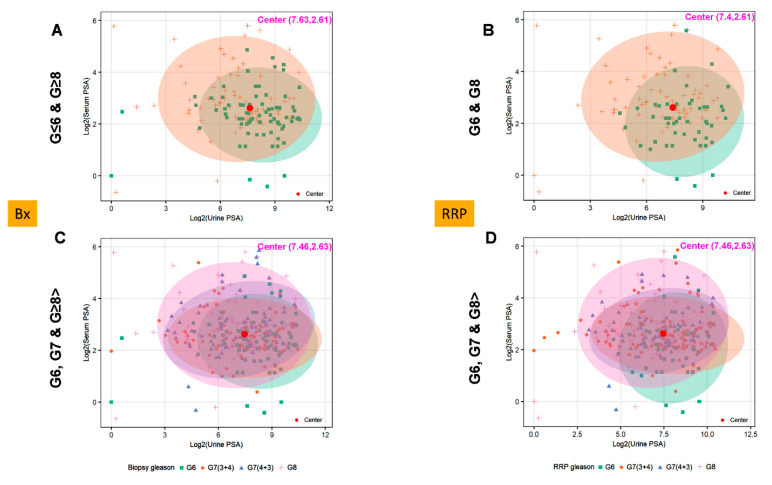
Utility of serum to urine PSA ratio in predicting the biopsy and pathological-based Gleason outcomes: The two-dimensional ellipse plots represent the distribution of PSA values in urine (*x*-axis) and serum (*y*-axis) for the biopsy (**A**) and radical prostatectomy (**B**) in GS ≤ 6 vs. GS ≥ 8 samples (**A**,**B**). The ellipse plots show four different ellipse distributions of Gleason scores (GS 6, 3 + 4, 4 + 3, and 8 > ) for surgical or pathological Gleason (**C**,**D**). The box and whisker plots showing the median PSA values and the overall dispersion of individual PSA values for serum (**E**) or urine (**F**), as well as their respective ratios in samples with surgical GS ≤ 6 vs. GS ≥ 8 (**E**,**F**,**G**). The box plots were further used to stratify the serum and urine PSA (**H**,**I**) and the ratio of serum to urine PSA (**J**) across different surgical grades (biopsy), along with their *p*-values.

**Figure 5 cancers-15-00960-f005:**
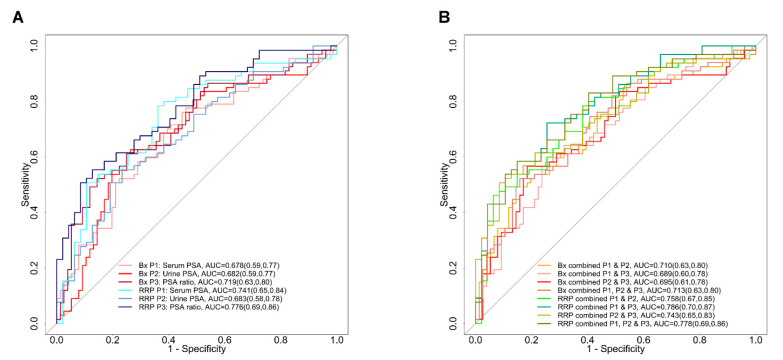
Performance of serum and urine PSA alone and with ratios across different Gleason scores. The ROC curves for the biopsy (Bx) or radical prostatectomy (RRP) serum or urine PSA to predict the AG disease. Comparing the urine to serum PSA and the ratio of serum to urine PSA demonstrated that serum to urine PSA ratios had the best performance in terms of AUC analysis, i.e., AUC = 0.719 and AUC = 0.776 in Bx and RRP specimens, respectively. Their combinatory approach (**B**) demonstrated that none of the combinations of the ROC curves between serum or urine PSA assays were superior to the ROC curves that were constructed for the ratio between serum to urine alone. Numbers in parentheses represent the 95% (CI) confidence intervals for each group (**A**,**B**).

## Data Availability

The data that support the findings of this study are not publicly available to ensure that the privacy of the research participants is not compromised. However, the data are available from the corresponding author upon reasonable request.

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
