# Peer review of "Urinary PSA and Serum PSA for Aggressive Prostate Cancer Detection"

_cancers, 2023, doi:10.3390/cancers15030960_

Round 1

Reviewer 1 Report

This interesting paper aims to support the utility of urine-based approaches for PSA assay to complement the current serum PSA and to serve as a proxy for prostate tissue-related expression levels. The manuscript needs a syntax check and it is lacking in several points that would add value to the entire manuscript: a major revision is required. 

·      Please lines 73-78 should be included in the methods. The introduction should only contain background and study aims.

·      Lines 78-80 should be deleted and included in the conclusion.

·      Please correct the typo in line 40.

·      The sentence in lines 45-46 “The production of PSA per cell and per gram of tissue is lower in prostate cancer than in 45 normal tissue and benign prostate hyperplasia (BPH)” could lead to misinterpretation, please reformulate it.

·      Why did you present the number of patients of cohorts 1 and 2 in the methods? Those should be presented in the result.

·      Methods and Results must be restructured. Methods should describe what was done to answer the research question, describe how it was done, justify the experimental design, and explain how the results were analyzed. Results should include only the findings of your study in a sterile manner. The findings include data presented in tables, charts, graphs, and other figures. Please re-organize the sections. 

·      Since large-sized sample studies and longer-term follow-ups are needed to validate urinary biomarkers' utility, I strongly suggest including this interesting paper (PMID: 36294423; PMCID: PMC9605140; DOI: 10.3390/jcm11206102) which aims to identify urinary molecules acting as biomarkers of the radicality of treatment in patients with PCa. Authors measured urinary levels of 8-OHdG and 8-iso-PGF2α before and after RARP surgery which has proven to help predict radicality (and perhaps local recurrence) following surgery.

·      You have stratified PCa stratified according to the degree of disease, please include this citation (PMID: 35409187; PMCID: PMC8999129; DOI: 10.3390/ijms23073826) where the genetic signature of autophagy has been found to have a potential means to stratify prostate cancer aggressiveness.

·      Please extrapolate from the discussion a conclusion section.

·      Check typos.

Author Response

Dear Dr. Kohar,

First of all, thank you very much for giving us this opportunity to revise our manuscript. We are now delighted to submit our revised manuscript. In this revised version we have added all of our reviewers’ suggestions and have proof read our manuscript for any typos. Below we have responded point-by-point to each of the reviewer’s comments.  The reviewer’s points are typed in bold, underlined and italicized for easier reference. Once again, we are very much thankful to you and to our reviewers for their insightful comments and suggestions.  We look forward to hearing from you. Sincerely, Naseruddin Höti (PhD)  

Reviewer I:

Comments and Suggestions for Authors

This interesting paper aims to support the utility of urine-based approaches for PSA assay to complement the current serum PSA and to serve as a proxy for prostate tissue-related expression levels. The manuscript needs a syntax check and it is lacking in several points that would add value to the entire manuscript: a major revision is required. 

Please lines 73-78 should be included in the methods. The introduction should only contain background and study aims.

We agreed with the reviewer suggestions, we have removed lines 73 to 78 from the introduction section and included them in our method and material section.

Lines 78-80 should be deleted and included in the conclusion.

We agree with our reviewer, in the revised version, we have deleted those lines from the introduction section and have included them in the conclusion section of our revised manuscript.

Please correct the typo in line 40.

We are thankful to our reviewer from his/her careful reading of our manuscript. We have corrected the typo in line 40 and have proof read our revised manuscript for any additional typos.

The sentence in lines 45-46 “The production of PSA per cell and per gram of tissue is lower in prostate cancer than in 45 normal tissue and benign prostate hyperplasia (BPH)” could lead to misinterpretation, please reformulate it.

We are very much thankful to our reviewer we have now re-worded the sentence to

“Several studies have demonstrated lower levels of PSA production in cancerous tissues” to avoid any confusion to our readers

Why did you present the number of patients of cohorts 1 and 2 in the methods? Those should be presented in the result.

We appreciate our reviewer’s comments, we have moved the numbers for the patients into our results section.

Methods and Results must be restructured. Methods should describe what was done to answer the research question, describe how it was done, justify the experimental design, and explain how the results were analyzed. Results should include only the findings of your study in a sterile manner. The findings include data presented in tables, charts, graphs, and other figures. Please re-organize the sections. 

We totally agree with our reviewer and appreciate his/her suggestions to help improve our article. We have now re-organized our methods and results section. We have removed part of the materials from the method section into our results section. Similarly, we have removed materials from results section into the discussion section.

  Since large-sized sample studies and longer-term follow-ups are needed to validate urinary biomarkers' utility, I strongly suggest including this interesting paper (PMID: 36294423; PMCID: PMC9605140; DOI: 10.3390/jcm11206102) which aims to identify urinary molecules acting as biomarkers of the radicality of treatment in patients with PCa. Authors measured urinary levels of 8-OHdG and 8-iso-PGF2α before and after RARP surgery which has proven to help predict radicality (and perhaps local recurrence) following surgery.

You have stratified PCa stratified according to the degree of disease, please include this citation (PMID: 35409187; PMCID: PMC8999129; DOI: 10.3390/ijms23073826) where the genetic signature of autophagy has been found to have a potential means to stratify prostate cancer aggressiveness.

We appreciate our reviewer for directing us toward these important manuscripts. We have now discussed these papers in the introduction section as well as in the discussion section of our revised manuscript.

Please extrapolate from the discussion a conclusion section.

We have included a conclusion by extrapolating it from the discussion section into our revised manuscript.

Check typos.

We have read the manuscript by ourselves and have additionally asked a native speaker to go over the revised version of our manuscript for any typos.

Reviewer 2 Report

In this manuscript, the authors applied urine PSA test to PCa patients’ samples for aggressive prostate cancer detecting. Their data provide a potential clinical utility to use urine PSA in combination with serum PSA for aggressive prostate cancer diagnosis. Most of the conclusions are significantly supported by their data. Most of the methods are appropriately described.

I have one major concern. For cohort 1, according to your description in main text and presentation in Figure 1, there are totally 146 samples, including NAG (G6) = 70 and AG (G8+G9+G10) = 76. But turning to main text 3.4 and Figure 3, the sample numbers for G8 are much more than the numbers for G8 in Figure 1. Do you mean G8 in this part is equal to AG? It makes readers confused. I understand you would like to present the sense of serum and urine PSA ratio in discriminating AG from NAG, but why do you use G6 vs G8 instead of NAG vs AG?

Here are some minor issues.

1.     Please organize your definition for NAG and AG into a unified description (P86-87 and Figure 1). Indeed, GS≥8 is equal to GS>7, and GS≤6 is the same with GS=6 in your case, but these make readers confused.

2.     P87-88, the definition for NAG and AG in Cohort 2 is lost.

3.     P131, please use the full name when you mention AUC at the first time.

4.     P243, using the serum to urine PSA ratio, urine is lost.

5.     P267-271, you didn’t cite Figure 3E and G in figure legend.

6.     P308, x-axis is urine while y-axis is serum.

One suggestion. There are some urine tests already existed for prostate cancer, such as MPS Test, developed at the University of Michigan, which combines serum PSA with urine T2:ERG, and urine PCA3. Have you ever considered combine urine PSA test with other biomarkers for prostate cancer existed in urine?

Author Response

Reviewer 2:

Comments and Suggestions for Authors

In this manuscript, the authors applied urine PSA test to PCa patients’ samples for aggressive prostate cancer detecting. Their data provide a potential clinical utility to use urine PSA in combination with serum PSA for aggressive prostate cancer diagnosis. Most of the conclusions are significantly supported by their data. Most of the methods are appropriately described.

I have one major concern. For cohort 1, according to your description in main text and presentation in Figure 1, there are totally 146 samples, including NAG (G6) = 70 and AG (G8+G9+G10) = 76. But turning to main text 3.4 and Figure 3, the sample numbers for G8 are much more than the numbers for G8 in Figure 1. Do you mean G8 in this part is equal to AG? It makes readers confused. I understand you would like to present the sense of serum and urine PSA ratio in discriminating AG from NAG, but why do you use G6 vs G8 instead of NAG vs AG?

We very much appreciate our reviewer suggestions and his/her understanding of our manuscript.   We have now carefully stated the Gleason scores for aggressive samples as (GS ≥ 8) which include all of the GS 8, 9 and 10. Similarly, for the nonaggressive (NAG) samples we have abbreviated the GS=6 to GS ≤ 6 for cohort 1. We have also added a detailed description of cohort 1 and cohort 2 into the introduction section to avoid any confusion to our readers.

Also, we very much liked our reviewer suggestions to simply state the AG and NAG when comparing G6 (NAG) vs G≥7 (AG) to avoid confusion among readers. In our revised manuscript we have incorporated these annotations in the text where appropriate.   

Here are some minor issues.

  1. Please organize your definition for NAG and AG into a unified description (P86-87 and Figure 1). Indeed, GS≥8 is equal to GS>7, and GS≤6 is the same with GS=6 in your case, but these make readers confused.

We once again are very much thankful to our reviewer. As per his/her suggestions we have defined the NAG and AG in our materials and methods section under the Urine samples and PSA assay as:

“The post-digital rectal examination (DRE) urine samples for cohort 1 comprised of distinct patient population with Gleason scores six or below (GS ≤6) that were classified as non-aggressive (NAG) and samples that were from patients with pathological Gleason score of eight or above (GS ≥8) that were aggressive patients (AG). A second group of samples that was cohort 2, comprised of even larger sample size which also included aggressive patients with Gleason scores of 7 that is (GS 3+4) and (GS 4+3) and were classified as aggressive patients (AG).”  

  1. P87-88, the definition for NAG and AG in Cohort 2 is lost.

Thank you once again to our reviewer for the careful reading of our manuscript. We have now added the definitions of NAG and AG for the Cohort 2 in our revised manuscript and have provided a description of AG and NAG in the methods and material section of our revised manuscript.

  1. P131, please use the full name when you mention AUC at the first time.

We have defined the AUC as Area under the cure in the text of our revised manuscript

  1. P243, using the serum to urine PSA ratio, urine is lost.

We have added  “Urine” to the serum to Urine PSA ratio in our revised manuscript.

  1. P267-271, you didn’t cite Figure 3E and G in figure legend.

Thank you once again to our reviewer for pointing that to our attention. We have now included the description of figure 3E and G in the figures and legends of our revised manuscript.

  1. P308, x-axis is urine while y-axis is serum.

We have corrected the x-axis for urine and y-axis for serum in the text of our revised manuscript.

One suggestion. There are some urine tests already existed for prostate cancer, such as MPS Test, developed at the University of Michigan, which combines serum PSA with urine T2:ERG, and urine PCA3. Have you ever considered combine urine PSA test with other biomarkers for prostate cancer existed in urine?

We are very much thankful to our reviewer for pointing us to the elegant work of the Michigan group on the urinary biomarker. We have cited two papers from the group in our revised manuscript.

It will be really interesting to evaluate urinary PSA with T2:ERG and PCA3 in our future studies and especially to evaluate the discriminating power of these markers in combinatory approaches in biopsied GS 7 patients i-e  GS 3+7 vs GS 4+3.

Round 2

Reviewer 1 Report

The manuscript has been deeply improved in structure and It is worthy of interest.